# OpenReview forum: "GRE Score: Generative Risk Evaluation for Large Language Models"
_NeurIPS.cc/2024/Workshop/SafeGenAi — SafeGenAi Poster_

### Official Review · Reviewer_gBYQ · 2024-10-09
**A useful metric and comprehensive study**

**Rating:** 7
**Confidence:** 4

**Review:**

In this paper, the authors propose an attack-independent evaluation metric. Based on this metric, they conduct comprehensive experiments regarding the safety, ethics, fairness, privacy, robustness of different LLMs. From my perspective, this metric will be useful for the community.

---

### Official Review · Reviewer_HDDu · 2024-10-10

**Rating:** 6
**Confidence:** 4

**Review:**

Summary

This paper deals with generative risk evaluation for LLMs. The authors propose a GRE score to quantify the robustness of LLMs in an attack-agnostic manner. This GRE score considers safety, privacy, robustness, ethics, and fairness. The evaluation displays high correlations with RTA metrics under red teaming attacks and is validated with lower computational cost.

Pros
- The paper is well written with clear motivation and idea presentation.
- The computation-friendly evaluation is significant.
- The theoretical guarantee is provided for the local GRE score as a robustness certificate.

Cons
- The motivation for choosing TAP as the red-teaming attack is not clear. Why not other attack methods that can also be adapted to the multi-dimensional evaluation, such as GCG, and AutoDAN? The only evaluation on TAP could make it less convincing.
- How does the GRE score align with human evaluation or GPT-4 evaluation? The paper involved a Longformer and a PLM classifier as the judgment classifier, which might not be convincing enough. Potential human evaluation (or at least GPT-4 evaluation) could make it more solid.
- Given that the GRE score calculation includes sampling latent variable z for paraphrasing, the stability needs to be analyzed. For example, what is the variance of GRE scores using 10 different random seeds? The practical parameters should also be determinative to reproduce the evaluation score.

---

### Official Review · Reviewer_q9SV · 2024-10-10
**The idea of quantitatively assessing the robustness of LLM is interesting, however, the question remains whether it is feasible to accurately quantify the overall robustness of LLMs based on such a narrow subset of prompts.**

**Rating:** 5
**Confidence:** 5

**Review:**

This paper introduces GRE scores, a novel conditional robustness certificate for large language models (LLMs). The proposed framework involves three key steps. First, synthetic samples are generated by loading risk-specific prompts that target selected risk evaluation tasks. Second, these generated prompts are input into an LLM to obtain responses and refusal predictions. Finally, the statistics from these responses are utilized to compute the GRE score, providing a measure of the model’s robustness under specified conditions.

The main contribution of this work lies in its attempt to quantitatively assess the robustness of LLMs. However, a key concern arises regarding the generalization of these findings, given the limited number of prompts and scenarios used in the evaluation. Compared to the wide variety of inputs typically encountered by LLMs, the number of risk-specific prompts employed here is relatively small. While it is possible to expose vulnerabilities and risks in LLMs using such carefully curated cases, the question remains whether it is feasible to accurately quantify the overall robustness of LLMs based on such a narrow subset of prompts.